# In Vitro Characterization of a Novel Human Acellular Dermal Matrix (BellaCell HD) for Breast Reconstruction

**DOI:** 10.3390/bioengineering7020039

**Published:** 2020-04-28

**Authors:** Sun-Young Nam, Dayoung Youn, Gyeong Hoe Kim, Ji Hwa Chai, Hyang Ran Lim, Hong Hee Jung, Chan Yeong Heo

**Affiliations:** 1Department of Plastic & Reconstructive Surgery, Seoul National University Bundang Hospital, Seongnam 13620, Korea; 99261@snubh.org (S.-Y.N.); dyoun@sookmyung.ac.kr (D.Y.); 2Department of Plastic and Reconstructive Surgery, Seoul National University Hospital, Seoul National University College of Medicine, Seoul 03080, Korea; hwe0216@gmail.com; 3The Institute of Tissue Engineering, HansBiomed Co. LTD., Daejeon 34054, Korea; cjh@hansbiomed.com (J.H.C.); hrlim@hansbiomed.com (H.R.L.); junghh@hansbiomed.com (H.H.J.)

**Keywords:** acellular dermal matrix, BellaCell HD, DermACELL, AlloDerm RTU

## Abstract

In the past, acellular dermal matrices (ADMs) have been used in implant-based breast reconstruction. Various factors affect the clinical performance of ADMs since there is a lack of systematic characterization of ADM tissues. This study used BellaCell HD and compared it to two commercially available ADMs—AlloDerm Ready to Use (RTU) and DermACELL—under in vitro settings. Every ADM was characterized to examine compatibility through cell cytotoxicity, proliferation, and physical features like tensile strength, stiffness, and the suture tensile strength. The BellaCell HD displayed complete decellularization in comparison with the other two ADMs. Several fibroblasts grew in the BellaCell HD with no cytotoxicity. The proliferation level of fibroblasts in the BellaCell HD was higher, compared to the AlloDerm RTU and DermACELL, after 7 and 14 days. The BellaCell HD had a load value of 444.94 N, 22.44 tensile strength, and 118.41% elongation ratio, and they were higher than in the other two ADMs. There was no significant discrepancy in the findings of stiffness evaluation and suture retention strength test. The study had some limitations because there were many other more factors useful in ADM’s testing. In the study, BellaCell HD showed complete decellularization, high biocompatibility, low cytotoxicity, high tensile strength, high elongation, and high suture retention strengths. These characteristics make BellaCell HD a suitable tissue for adequate and safe use in implant-based breast reconstruction in humans.

## 1. Introduction

An acellular dermal matrix (ADM) is a form of biological skin that is extracted from cadaveric skin in processes like decellularization and terminal sterilization [1]. It mainly constitutes fibrillary collagen, elastin, glycoproteins, proteoglycan, glycosaminoglycan, growth factors, and basement membrane. Elastin and collagen facilitate the tensile strength, while elasticity, proteoglycans, and laminin help in the induction of angiogenesis and binding to the connective tissues [1]. Moreover, growth factors control cell behavior, and cross-linked ADM degrades and releases biochemical signals at similar rates to that of native tissue extracellular matrix (ECM) [2]. When ADM is implanted into a body part, it influences host remodeling responses like cell movement, proliferation, and differentiation, and works an inductive support for the formation of site-specific useful host tissues.

Recently, ADM has been found to be useful in several fields like abdominal wall surgery, cleft palate repair, nasal septal reconstruction, breast reconstruction, and the evidence for its uses are evolving and developing. There are two major methods for breast reconstruction, namely implant-based reconstruction and autologous tissue-based reconstruction [3]. Due to the improved detection of breast cancer there is a demand for breast reconstruction development in breast manufacturing methods. Expander-based reconstruction has no donor indisposition, they are simple to conduct, and has become much common. The most popular technique is the placement of the expander below the pectoralis major muscle [4]. The fan-shaped pectoralis muscle does not cover the inferolateral aspect of the breast implant, and the ADM is implanted as a sling in between the pectoralis muscle and the inframammary fold [5]. The benefits of using the ADMs are the support provided to the inferolateral, greater fill volume, the greater definition of the inframammary pleat, and lesser capsular contracture. ADM is an indispensable biomaterial in breast reconstruction, through the refinement of the surgical approach and the manufacturing process.

AlloDerm was developed in 1994; different ADM products for breast reconstruction are currently available in the market. Every material is derived from a specific source and placed in a variety of processing procedures, sterilization, and storage environment. A human-derived ADM, BellaCell HD, was recently developed by Biomed Corporation, through a unique process. For the biomaterials to be applied in living organisms, the in vitro studies of their stability and biocompatibility should be performed [6]. This research is an in vitro study examining the decellularization status, biocompatibility, and mechanical features of BellaCell HD.

## 2. Materials and Methods 

### 2.1. ADM_S_

The research used BellaCell HD by Hans Biomed Corporation (Daejeon, Korea) and two human ADMS for comparison, i.e., AlloDerm Ready to Use (RTU) by LifeCell Corp. (Branchburg, NJ, USA) and DermACELL by LifeNet Health (Virginia Beach, VA, USA). Below are the processes that were conducted on the three human ADMS for evaluation of the BellaCell HD. The BellaCell HD is a novel human ADM.

### 2.2. Decellularization Assessment

For the study, paraffin-embedded ADMs were sectioned at 5 μm thickness, and after removing the paraffin, the sections were rehydrated with a reducing series of alcohol concentration to water. The standard protocol for H & E was followed. The samples were evaluated by light microscopy at different range magnifications to inspect the presence of the cells and collagen fibers. The light microscope was 40, 100, and 200 magnifications.

### 2.3. Cell Culture

For experimental purposes, the ADM specimens were divided into 1 by 1 cm pieces, put into 24-well cell culture dishes, washed with PBS twice, and later incubated in media at 37 °C and 5% carbon dioxide for ten minutes, before the cell seeding process. NIH3T3 and L-929 mouse fibroblasts were grown in Dulbecco’s modified Eagle’s medium added with 100 µg/mL streptomycin, 100 U/mL penicillin, and 10% heat-inactivated fetal bovine serum. The cells were cultured at 37 °C in 5% CO_2_/95% air.

### 2.4. Proliferation Assay

For cell proliferation assay, NIH3T3 fibroblasts were seeded on the ADMs at 5 × 10^4^ cells/mL and incubated for 1, 7, and 14 days. Cell proliferation was assessed with the use of an MTT assay. MTT solution at the 100 µL/well was added and incubated for 4 hours at 37 °C. The formazan crystals were liquified in dimethyl sulfoxide (DMSO), and the optical densities were calculated at 570 nm by an ELISA reader.

### 2.5. Cytotoxicity Assay

Cytotoxicity assays with the cultured cells were used for the testing chemicals and drug screening. For this study, sterile physiological saline were added and eluted at 37 ± 2 °C for 72 ± 2 h. Minimum essential medium (1×) with Earle’s salts (MEM-E, Flow Labs., Rockville, MD, USA) was used as a negative control after cell exposure to a similar environment with the test sample and DMSO as a positive control, and sterile physiological saline as the solvent control. The eluate was centrifuged at 3000 rpm for 5 min, and the supernatant was used as the test group. L-929 fibroblasts were seeded at a concentration of 5 × 10^4^ cells/well and incubated for 24 hours in a 37 °C saturated incubator with 5% CO_2_. The medium was detached after one day, and the eluate from the negative control group, the positive, the solvent regulator cluster, and the test assembly was replaced with a medium mixed in the same amount, with a 2-fold content of the MEM medium. After incubation for two days at 37 °C in a humidified incubator with 5% CO_2_, cell morphology was examined under a light microscope at 200× magnification, and cell viability was evaluated by MTT assay.

### 2.6. Uniaxial Tensile Testing

Biological soft tissue is a non-linear material, and the mechanical features affect the quality of life [7]. For this research, seven specimens (n = 7) measuring 10 mm × 7 mm were prepared from each of the ADMs (based on the American Society for Testing and Materials (ASTM) specification #D638-03). The average thickness of the specimens for each ADM was 1.97 mm for BellaCell HD, 1.3 mm for AlloDerm RTU, and 1.9 mm for DermACELL. The sample used in this study had different thickness. Each sample was oriented vertically in the Instron material testing system with a 3.0 cm gauge length, and approximately 2.0 cm of the specimen was fixed firmly in each pneumatic grip. Samples were pulled to a uniaxial pressure at a rate of 30 mm/min until failure. When the ADM was broken, the elongation ratio (%), the maximum length divided by original length, and the maximum load (N) was determined. Tensile strength was evaluated by the division of maximum load with the cross-sectional area (mm^2^) of the sample to yield the value in megapascal (MPa) units, and 1 N/mm^2^ equaled 1 MPa.

### 2.7. Stiffness Testing

Most of the biomaterials were viscoelastic, showing time dependence in the cell’s response to the loads. In this study, stiffness testing was conducted using a custom test fixture. The custom test fixture was fabricated based on American Society for Testing and Materials (ASTM) specification #F1306. Three specimens (BellaCell HD n = 3, AlloDerm RTU n = 3, DermACELL n = 3) measuring 5 cm × 5 cm were prepared. The average thickness of each ADM specimen was 1.88 mm for BellaCell HD, 1.16 mm for AlloDerm RTU, and 1.3 mm for DermACELL. The sample used in this study had different thickness. The specimen was fixed between the upper and lower jigs, and the probe moved downward to compress the sample at a degree of 25 mm/min. The stiffness was evaluated through division of the load sustained with sample (N) in the stiffness examination by the movement (mm) of the probe.

### 2.8. Suture Retention Strength Testing

The cell width and the distance of the joint bite from the sample free edge are the most significant geometrical parameters [8]. For this work, four samples (2 cm × 4 cm) were prepared from each ADM. From the bottom of the sample, a suture was passed through the ADMs (1.0 cm). A pulling rate of 20 mm/min was applied as the suture tore out of the ADMs. The maximum load (N) was recorded as mean ± standard error of the mean (SEM).

### 2.9. Statistical Analysis

All figures are reported as mean ± SEM. Statistical analyses were conducted using SPSS statistical software, and it was beneficial in providing the frequencies and bivariate statistics. For all data, significant differences were established using an unpaired *t*-test. For all analyses, *p* < 0.05 was explained as statistically substantial.

## 3. Results

### 3.1. Decellularization Assessment

BellaCell HD and DermACELL displayed complete decellularization under a light microscope after staining. Complete decellularization showed the compatibility of the BellaCell HD for use in breast reconstruction. Some cellular debris between collagen fibers were visible on the AlloDerm RTU, as shown (Figure 1).

### 3.2. Biocompatibility Assessment

Cell proliferation in BellaCell HD was highest in the one day, though not statistically significant. On the 7 and 14 days, BellaCell HD was higher than in the AlloDerm RTU (Figure 2A). There was no significant difference in cell proliferation between BellaCell HD and DermACELL. After the 1, 7, and 14 days of incubation with NH3T3 fibroblasts, invasive cells on the ADM’s surface were photographed with the use of a light microscope at 100× magnification. All samples displayed the same degree of cell adhesion on the first day, as demonstrated (Figure 2B). Many cells overgrew in the BellaCell HD in comparison to the other matrices. Cell proliferation was not constant in the AlloDerm RTU, but growth was mainly seen on the dent surface (Figure 2B). 

### 3.3. Cytotoxicity Assay

MTT assay showed that all three products had a cell viability of over 90%, indicating no cytotoxicity as illustrated in Figure 3A. In the cytotoxicity assay, a significant decline in cell count was observed in the positive control using DMSO. There was a rise in cell count in the negative control, as seen under a light microscope at 200× magnification (Figure 3B). In all test groups using BellaCell HD, AlloDerm RTU, and DermACELL, there was a growth with a spindle-like structure (Figure 3B). 

### 3.4. Uniaxial Tensile Test

Uniaxial tensile testing showed that the maximum load at the ADM break was 444.94 N for BellaCell HD, which was higher than that for AlloDerm RTU (181.92 N), and marginally lower for DermACELL (492.11 N), which was not statistically significant (Figure 4A). The tensile strength of BellaCell HD was 22.44 MPa. It was significantly higher than the 14.34 MPa observed for AlloDerm RTU and not substantially different from the 26.12 MPa observed for DermACELL (Figure 4B). The elongation ratio at the ADM break was 118.41% for BellaCell HD, 126.38% for AlloDerm RTU, and 104.13% for DermACELL (Figure 4C).

### 3.5. Stiffness Testing

A biomaterial requires mechanical strength to uphold integrity until full regeneration of the tissue and has to sustain space for the cell ingrowths and nutrient absorption in vitro, thereby supporting the loadings in vivo [9]. The scaffold should be able to match its characteristics to those of initial tissues to avoid pressure shielding and offer the cell appropriate mechanical cues [10]. The stiffness testing showed that DermACELL had the highest stiffness of 0.90 N/mm, while those of AlloDerm RTU and BellaCell HD were measured 0.28 N/mm and 0.44 N/mm, respectively, as illustrated in Figure 5.

### 3.6. Suture Retenstion Strength Testing

The ADM must withstand tearing at the point of the suture when tension pulls at the suture. Break starting strength is firm against the assessment parameter discrepancies, and it is dependent on the geometry of the sample [11]. The contrast of suture preservation and mode one crack opening evaluations shows the linear relationship between break starting and tearing energy [12]. The results of suture retention strength testing showed that the maximum load for BellaCell HD was 97.06 N, which was higher than that for AlloDerm RTU and for DermACELL, as shown in Figure 6. However, these differences were not statistically significant (*p* > 0.05).

## 4. Discussion

The present study used the novel ADM, BellaCell HD, in an in vitro environment, contrasting it with two commercially available human ADMs. The aim was for the ADM to be successfully utilized in expander-based breast reconstruction. The BellaCell HD showed complete decellularization, viewed using H & E staining under a light microscope [13]. The BellaCell also showed high cell adhesion and cell proliferation without cytotoxicity in the biocompatibility assessment. The BellaCell HD displayed a high tensile strength, elongation, low stiffness, and high suture retention power in the mechanical property assessment. 

The manufacturing process of ADMs consists of decellularization, preservation, and sterilization stages. The most important phase is decellularization using physical, chemical, or biological techniques [6]. Manufacturing of each product is done differently. The purpose of the decellularization process is the removal of antigenic material while preserving extracellular matrix biochemistry and structure [14]. The presence of residual DNA in the biological scaffold materials results in an inflammatory response. Previous research found that the presence of cells within a biomaterial is linked with increased macrophage M1 polarization, increased proinflammatory cytokines, and weak remodeling results in a primate model [15]. Reasons for complication development in the ADMs in breast reconstruction are multifactorial [16]. Hence, this research holds that BellaCell HD is immunologically safe for implantation in the human body.

Immune cells like lymphocytes, granulocytes, macrophages and mast cells, fibroblast, and myofibroblasts recolonize the original ADM during implantation in the breast reconstruction process [14]. Capsule and capillaries are formed by the fibrosis and neovascularization [17]. It should be compatible and capable of inducing the biological responses like host cell adhesion and cell proliferation with no cytotoxicity to the host tissue. Lack of biocompatibility leads to an imbalance resulting in implant mobility that causes infection, reconstructive failure, and seroma. High compatibility of BellaCell HD might result in a favorable outcome when employed in breast reconstruction.

The benefit of using ADM in expander-based breast reconstruction is because it offers physical support to the implant hence preventing shifting or bottoming out [18]. High tensile strength ADM is a requirement for breast implantation [18]. Using a low tensile ADM the tissue would be vulnerable to matrix rupture, which could result in implant malposition [19]. After implantation, the expander was inflated for some months to get an adequate amount of skin, similar to that in the contralateral breasts. Therefore, ADM should have sufficient tensile strength to withstand the inflating pressure of the expander [18]. BellaCell HD showed a high tensile strength; thus; it was capable of providing sufficient physical support when used in breast reconstruction.

The paramount result of this research was that BellaCell HD showed both a high tensile strength and elongation ratio, which meant elasticity and flexibility, respectively. The AlloDerm RTU had low stiffness rate, and lower tensile strength [1]. The tensile power strength of DermACELL was high, like that of BellaCell HD, though with high stiffness. The elements helped the surgeons in handling ADM efficiently and overcoming the size discrepancy between the standardized ADM material and the spaces that required coverage for every patient [20]. Most surgeons create a vertical or horizontal stab incision to the ADM, whereas others mesh the ADM using a skin graft Mesher [21]. However, all of these techniques have the shortcoming of increasing the contact area between the implant and the mastectomy flap [22]. Therefore, the high elasticity and pliability of the BellaCell HD would help in bridging the gap, particularly by stretching it. Currently, prepectoral implant placement with complete coverage by ADM has become popular because of low postoperative pain and low animation deformity [23]. Since more complex techniques are needed for complete wrapping with ADM, the high flexibility of the ADM will help safe handling.

In the suture retention strength test, BellaCell HD had the uppermost suture retention strength of the three human ADMs examined. When the ADM was implanted as a hammock shape in implant/expander-based breast reconstruction, ADM was sutured with the elevated pectoralis major superior and the chest wall at the inframammary and lateral mammary fold inferolateral [24]. Although few human research studies have been conducted, it is speculated that the suture site wound is more vulnerable to dehiscence than the ADM itself, until it is fully integrated with the host tissue [25]. Therefore, the high suture retention strength of BellaCell HD will aid in preventing implant herniation through the suture site. For BellaCell HD to expand its indications like other common ADMs, it must have appropriate suture retention strength that can withstand high tensions such as abdominal walls, which is supported by these findings.

Indeed, AlloDerm RTU and DermACELL have been used in clinical applications such as breast reconstruction. DermACELL is an appropriate adjunct to the post-mastectomy outcome of reconstruction with expanders. Histological observation showed early graft integration at 6 weeks post-implantation [26]. Clinical data on AlloDerm RTU used in breast reconstruction showed full incorporation and integration into the host tissue [27]. Based on the present study, BellaCell might also reduce complications that lead to reconstruction failure.

The limitation of this study is that biocompatibility and mechanical properties of ADM were tested only in an in vitro setting. We used only NIH3T3 and L-929 mouse fibroblasts to assess cell proliferation and cell viability. However, not only fibroblasts but also myofibroblast, lymphocytes, macrophages, granulocytes, and mast cells were involved in the ADM integration [28]. Moreover, antibodies, complement, and cytokines also perform a significant part in the host response to ADM [29]. Therefore, to identify the exact mechanism through which ADM integrates into the human body and to use it to create an ideal ADM, further detailed studies are recommended. The ADM facilitates the biochemical change through a process called “stretching” after implantation, which varies from product to artefact [30]. However, it is challenging to manufacture individual multi-vector forces in an in vitro setting. In vivo studies would be required to address this, to help surgeons predict the need for an ADM sling overcorrection in implant/expander-based breast reconstruction.

Authors should discuss the results and how they can be interpreted in perspective of previous studies and of the working hypotheses. The findings and their implications should be discussed in the broadest context possible. Future research directions might also be highlighted.

## 5. Conclusions

The BellaCell HD showed high compatibility, low cytotoxicity, high tensile strength, and complete decellularization, thus, displaying its suitability for use in breast reconstruction.

## Figures and Tables

**Figure 1 bioengineering-07-00039-f001:**
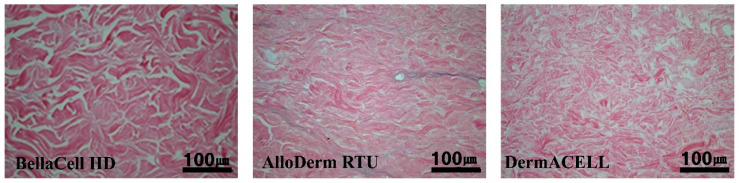
Decellularization assessment. The extracellular matrix of acellular dermal matrices (ADMs) was assessed by H&E staining (scale bar = 100 μm). The BellaCell HD and DermACELL display complete decellularization while the AlloDerm Ready to Use (RTU) shows cellular debris between the figures.

**Figure 2 bioengineering-07-00039-f002:**
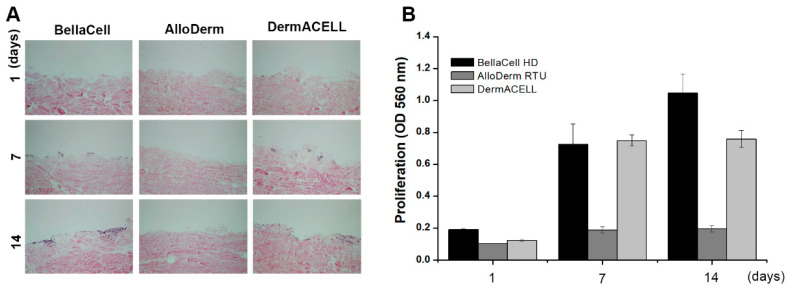
Biocompatibility assessment. (**A**) Cell adhesion on the surface of ADMs was observed under microscope after staining with H&E (**B**) Cell proliferation was evaluated by a 3-(4,5-dimethylthiazol-2-yl)-2,5-diphenyltetrazolium (MTT) assay. Cell adhesion and proliferation in a span of 1, 7, and 14 days. Figure 2A is a representation of cell adhesion, while Figure 2B shows the cell proliferation rate and incubation days. The cells overgrew with time in the BellaCell HD in comparison to the other two matrices, and the AlloDerm RTU cell growth was viewed at the dent. Each datum represents the ±SEM of three independent experiments.

**Figure 3 bioengineering-07-00039-f003:**
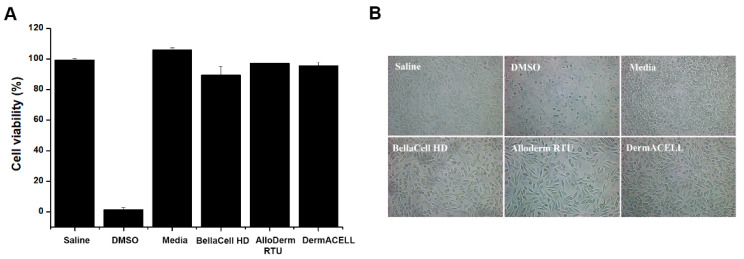
Cytotoxicity assay. (**A**) Cell viability was assessed by an MTT assay. (**B**) Cell morphology was observed under the light microscope at 200× magnification. Cytotoxicity of cell growth and cell viability in the three tissues. Figure 3A shows 90% cell viability and no cytotoxicity. In Figure 3B, cells grew in a spindle-like material. Each datum represents the ±SEM of three independent experiments.

**Figure 4 bioengineering-07-00039-f004:**
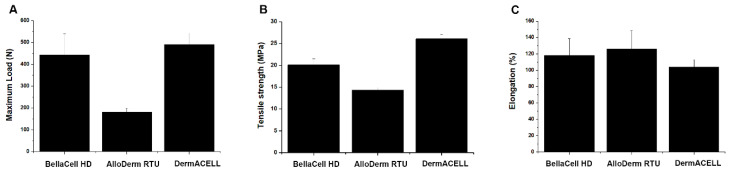
Uniaxial tensile test. (**A**) Maximal load, (**B**) tensile strength, and (**C**) elongation. Figure 4A shows the 444.94 N break for BellaCell HD, AlloDerm RTU 181.92 N, and 492.11N for DermACELL. The tensile for the BellaCell HD, Alloderm RTU, and DermACELL is 22.44 MPa, 14.34 MPa, and 26.12 MPa (Figure 4B). Figure 4C represents the elongation ratio of 118.41%, 126.38%, and 104.13% for BellaCell HD, Alloderm RTU, and DermACELL products, respectively. Each datum represents the ±SEM of three independent experiments.

**Figure 5 bioengineering-07-00039-f005:**
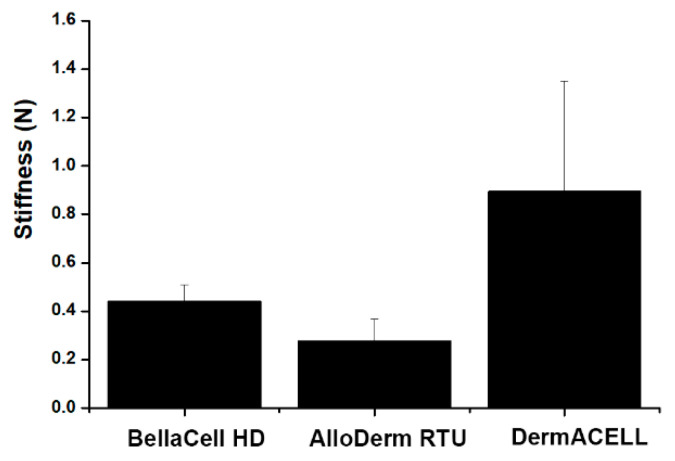
Stiffness testing. Stiffness testing was performed using a custom test fixture. The custom test fixture was fabricated on the basis of American Society for Testing and Materials (ASTM) specification #F1306. It displays a stiffness of 0.90 N/mm for DermACELL, 0.28 N/mm of AlloDerm RTU, and 0.44 N/mm for BellaCell. Each datum represents the ±SEM of three independent experiments.

**Figure 6 bioengineering-07-00039-f006:**
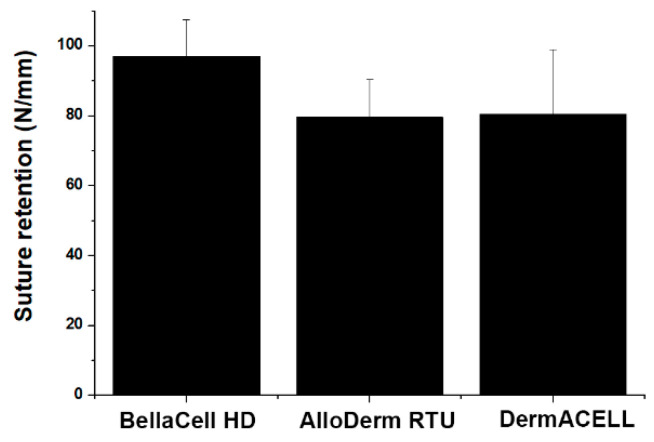
Suture retention strength test. Four samples (2 cm × 4 cm) were prepared from each ADM. From the bottom of the sample, a suture was passed through the ADM (1.0 cm). A pulling rate of 20 mm/min was applied as the suture tore out of the ADMs. The maximum load (N) was recorded as mean ± SEM. It displays the maximum load of 97.06 N, 79.65 N, and 80.48 N for BellaCell HD, AlloDerm RTU, and DermACELL, respectively. Each datum represents the ±SEM of three independent experiments.

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
