# Peer review of "In Vitro Characterization of a Novel Human Acellular Dermal Matrix (BellaCell HD) for Breast Reconstruction"

_bioengineering, 2020, doi:10.3390/bioengineering7020039_

Round 1

Reviewer 1 Report

in Abstract the word DermACELL is spelled incorrectly.

word missing line 251: the most XXX phase is decellularization using physical...

very nice paper but the conclusion is WILDLY over-drawn.  this is an in vitro preclinical paper.  The next step is NOT to conclude safe use in humans!! you must recommend further preclinical studies that address your limitations (such as lack of cytokine testing or other inflammatory markers), and then suggest possible animal model, NOT conclude safety for human use.

Author Response

Point 1: in Abstract the word DermACELL is spelled incorrectly.

Response 1: It has been corrected in page 1 line 18.

Point 2: word missing line 251: the most XXX phase is decellularization using physical...

Response 2: It has been corrected in page 7 lines 254-255.

Point 3: very nice paper but the conclusion is WILDLY over-drawn.  this is an in vitro preclinical paper.  The next step is NOT to conclude safe use in humans!! you must recommend further preclinical studies that address your limitations (such as lack of cytokine testing or other inflammatory markers), and then suggest possible animal model, NOT conclude safety for human use...

Response 3: It has been corrected in page 8 lines 302-307.

Reviewer 2 Report

I would encorage the Authors to do two things. First is to address that fact that the samples used for biomechanical testing had different thickness (same thickness should have been better). Next would be to (briefly) discuss the clinical evidence for each material.

Author Response

Point 1: I would encorage the Authors to do two things. First is to address that fact that the samples used for biomechanical testing had different thickness (same thickness should have been better).

Response 1: It has been added in page 3 lines 120-121 and line 136.

Point 2: Next would be to (briefly) discuss the clinical evidence for each material.

Response 2: It has been added in page 8lines 302-307 and Ref. 26 and Ref. 27.
